# Soil Sample Analysis of *Bacillus anthracis* Contaminated Animal Burial Sites

**DOI:** 10.3390/microorganisms12101944

**Published:** 2024-09-25

**Authors:** Mitat Sahin, Thomas R. Laws, Hugh Dyson, Ozgur Celebi, Mehmet Doganay, Fatih Buyuk, Les Baillie

**Affiliations:** 1Department of Microbiology, Faculty of Veterinary Medicine, Kafkas University, Kars 36300, Türkiye; mitats@hotmail.com (M.S.); fatihbyk08@hotmail.com (F.B.); 2Faculty of Veterinary Medicine, Kyrgyz-Turkish Manas University, Chingiz Aitmatov Campus, Djal, Bishkek 720038, Kyrgyzstan; 3Defence Science and Technology Laboratory, CBR Division, Porton Down, Salisbury SP4 0JQ, UK; trlaws@dstl.gov.uk (T.R.L.); ehdyson@dstl.gov.uk (H.D.); 4Department of Medical Microbiology, Faculty of Medicine, Kafkas University, Kars 36100, Türkiye; ozgurcelebi36@hotmail.com; 5Department of Infectious Diseases, Faculty of Medicine, Lokman Hekim University, Ankara 06530, Türkiye; mehmet.doganay@lokmanhekim.edu.tr; 6School of Pharmacy and Pharmaceutical Sciences, Cardiff University, Cardiff CF10 3NB, UK

**Keywords:** *Bacillus anthracis*, spore persistence, soil reservoirs, sample plan, environmental exposure, Kars region, endemic anthrax

## Abstract

Environmental contamination with *Bacillus anthracis* spores poses clear threats to livestock that play key roles in the economies of pastoral communities. Regular monitoring of contaminated sites is particularly important in anthrax-endemic parts of the world, such as Kars province in eastern Türkiye, where the Veterinary Microbiology Department of Kafkas University has conducted an anthrax surveillance programme for over 30 years. We reviewed the microbiological results of 232 soil samples collected during 2009–2023, from sites known to be contaminated with *B. anthracis* spores following burial or butchering of infected animal carcasses. Twenty-five contaminated sites in 16 villages were studied. Samples were taken from a total of 61 different positions within these sites and viable spores were detected in 136 (58.6%) of the samples examined. Of the 96 samples from which spores were not recovered, subsequent samples from the same positions proved positive on 21 occasions. Using a standardised sampling plan, it was discovered that samples taken 1–2 m on a downward slope from the centre-point of contamination had higher (*p* < 0.001) spore concentrations than those taken from other positions. Although spore concentrations at some sampling positions varied over time, the overall values remained stable. This finding contrasts with observations in other parts of the world where spore concentrations tend to decline with time and may reflect regional differences in soil composition that permit more prolonged spore persistence. Concentrations of >100 spores/g soil were found in 10 (66.7%) of the 15 samples taken 10–13 years following a contamination event. These results demonstrate the longevity of viable anthrax spores in the soil of agricultural environments following decomposition of infected animal carcasses, and therefore the need for prolonged bacteriological monitoring of contaminated sites. Furthermore, they underline the importance of appropriate decontamination, as burial on its own does not eliminate all spores.

## 1. Introduction

The natural reservoir of the pathogen *Bacillus anthracis* is soil. The bacterium is able to survive for long periods in soil due to its ability to form resistant spores that are able to infect grazing domestic and wild herbivores. Transmission to humans occurs primarily through close contact with the meat and products of infected animals [1].

Environmental contamination with *B. anthracis* spores poses clear threats to the health of livestock, particularly cattle and sheep, which play a key role in the economies of agricultural communities. Furthermore, in some nomadic pastoral cultures, cattle, goats and sheep constitute the main repository of family wealth, so their loss from disease can have a significant impact on future generations. Monitoring of contaminated sites is therefore particularly important in anthrax-endemic parts of the world, such as the Kars region in eastern Türkiye. The Veterinary Microbiology Department of Kafkas University in Kars has conducted an anthrax surveillance programme in this region for over 30 years and has undertaken planned surveys of known contaminated sites as well as analysis of samples routinely delivered to the department by farmers. Since 2008 veterinarians have carried out a sustained animal vaccination programme, accompanied by systematic education of livestock owners regarding the presenting features of anthrax and the risks of infection. Burial procedures for infected animal carcasses have also improved and these measures, together with the government compensation scheme that was introduced in 2012 [2], are believed to be responsible for the decrease in animal and human cases observed in the Kars region over the last decade.

It is well known that humans can contract anthrax following inhalation of airborne spores of *B. anthracis*. Historically, this has occurred in the context of occupational exposure to spore-contaminated animal fleeces in textile mills, where inhalation anthrax was known as Wool-sorter’s Disease. However, not all exposed individuals in these industrial environments develop clinical disease [3,4]. Indeed, studies of workers in a Belgian wool-processing factory [5] with previously documented environmental anthrax spore contamination [6], and in a New Hampshire goat-hair processing factory [3], found that individuals with no history of infection or vaccination can have detectable circulating antibodies against anthrax protective antigen (PA).

These studies led to the hypothesis proposed by Kissling et al. [5], amongst others, that healthy individuals living and working in naturally contaminated, non-industrial environments could also develop antibodies against anthrax toxin antigens as a consequence of subclinical exposure. The validity of this hypothesis was strengthened by a recent paper by Buyuk et al. [7], reporting the presence of anti-anthrax toxin IgG antibodies in the serum of healthy individuals living in the Kars region of eastern Türkiye, who had no history of previous clinical anthrax infection.

In an effort to estimate the level of *B. anthracis* spore exposure of individuals participating in the Buyuk study, we reviewed the microbiological analyses of soil samples collected from sites across the Kars region which were known to be contaminated following the disposal of infected animal carcases. In addition to determining the level of spore contamination at each site, when practical, we monitored the impact of location and time since contamination on spore viability.

## 2. Methods

### 2.1. Soil Sampling

Results are reported for all 232 soil samples taken during 2009–2023 from sites known to have been contaminated with *B. anthracis*; dates of contamination ranged from 2006 to 2023. Soil samples were derived from sites where an animal that had died of anthrax had been buried, or a carcass of an anthrax-infected animal had been butchered on open ground. Nearly all (>98%) the animals involved were cattle, the remainder being sheep. Samples were either collected during surveys planned by the Microbiology Department (n = 216) or taken from a known contaminated site by farmers and brought to the department for routine analysis (n = 16). These latter samples consisted of 0.5–1 kg of superficial soil from the centre of the contaminated site; they were not taken or handled in any systematic way before arrival at the laboratory. In contrast, samples obtained during planned surveys were taken in a standardised manner [8,9] with 0.1–0.5 kg of soil being collected from the top 10 cm of ground by a trowel that was decontaminated between each sampling. Where possible, multiple samples were collected from different positions at a given site to increase the likelihood of isolating *B. anthracis*, as it had previously been shown that the distribution of spores across a contaminated site was uneven [8,10]. Sixty-one sampling positions from 25 contaminated sites in 16 villages were studied altogether.

Sites where animal carcasses had been buried were relatively well defined, around 2 m by 2 m in extent, while areas with superficial contamination following butchering of an infected carcass on open ground were more irregular. An envelope sampling method [11] was employed at both types of sites for the Microbiology Department surveys, with 0.1–0.5 kg of soil being collected from the top 10 cm of ground at different positions within each contaminated site. One sample (number 1) was taken from the centre of a 1 m^2^ area within the site, without placing a metal peg in the carcass remains. As a general rule, a further four samples (numbers 2–5) were taken from positions round the periphery of the contaminated area [8], at a distance of 0.5 m along lines towards the four points of the compass (N, E, S, W). For sites with contamination areas less than 1 m^2^ in extent, either the distances between the four cardinal sampling points were reduced or fewer samples were taken. For large sites, up to five additional samples (numbered 6–10) were taken at intervals for a further 3 m down the slope (the maximum gradient at any site being <15°). Sampling positions were recorded on a map of the site (an example is shown in Figure 1A) to ensure that subsequent samples could be taken as closely as possible from the same places. All soil samples were stored in the laboratory at room temperature prior to culture.

The information collected for each soil sample included the following: location (district, village, site and position number), reason for sampling (planned survey or routine analysis), type of contamination (ground surface or animal burial), date of contamination and date of sampling. These details are listed together with the spore concentration (count/g soil) in Appendix A, and a map of Kars province with the administrative districts in which the villages are situated is shown in Appendix A.

The precision of recording contamination dates at individual sites varied. Due to lack of written records in the villages concerned, the dates of contamination for the earliest sites studied could only be identified to within a range of 1–2 years. Sampling dates were more precise, with nearly all being recorded to the nearest month. In order to avoid the risk of overestimating the longevity of viable spores in soil, the interval between contamination and sampling (Column I in Appendix A) was therefore calculated as the number of days between the *final* day of the period listed for the contamination date and the *first* day of the period listed for the sampling date.

### 2.2. Culture and Identification of B. anthracis

Following collection, all soil samples and isolated bacteria were processed in the biocontainment facilities at Kafkas University within BSL 2-Plus (A2 type) bio-safety cabinets (ESCO, SG). A portion of soil (40 g) from each sample was mixed with distilled water (200 mL), shaken vigorously by hand, and the resulting suspension left at room temperature for 30 min. A 1 mL aliquot was then taken from the suspension and subjected to 10-fold serial dilutions, which were heat treated at 62.5–63 °C for 20 min to inactivate vegetative bacteria. Following this, a 150 µL aliquot from each dilution was spread across the surface of a 7% (*v*/*v*) sheep blood agar (Thermo Fisher Scientific, Waltham, MA, USA) plate in duplicate and incubated overnight in air at 37 °C. The plates were then examined for the presence of colonies with characteristic *B. anthracis* morphology (rough appearance, non-hemolytic when cultured on sheep blood agar) [1]. The limit of detection was 13.2 cfu per gram of soil.

Suspected *B. anthracis* colonies were sub-cultured onto fresh sheep blood agar plates to determine sensitivity to diagnostic gamma phage and penicillin (10 units) (Oxoid, UK). The plates were then incubated overnight at 37 °C and examined for signs of inhibition (lysis and antibiotic inhibition, respectively). If these tests produced contradictory results, colonies were subjected to further phenotypic analysis which included Gram staining, determination of motility and capsule production when incubated in the presence of bicarbonate and carbon dioxide. Finally, pXO1 and pXO2 plasmid-based PCRs were carried out, using PA 5/8 for protective antigen and Cap 6/103 for capsule, respectively [12].

Degrees of genetic relatedness in selected isolates were determined by single nucleotide polymorphism (SNP) sub-typing, followed by high-resolution genotyping using 25-loci variable-number tandem repeat analysis (MLVA-25). This was carried out by the Lugar Center for Public Health Research at the National Center for Disease Control, Tbilisi, Georgia [13].

### 2.3. Statistical Analyses

Data (see Appendix A) were analysed using the software SPSS V27.1 (IBM, Armonk, NY, USA). Graphs were prepared using Graphpad PRISM V9.0. For analysis, bacterial count data were $\log_{10}y+1$ transformed to better fit a normal distribution. Linear mixed modelling was used to analyse these data. Only a main effect model was fitted, and no attempt was made to include interactions. The residuals from the final model were Gaussian (assessed by QQ plot), and there was no evidence for heteroscedasticity (assessed by residual plot). Data were analysed by a mixed model in which the experimental unit was held as the site. Sampling Position and Time since Contamination were used as explanatory variables. Village and Administrative District were nested and tested for effect in separate models. Descriptive statistics were calculated using Graphpad PRISM V9.0.

## 3. Results

Throughout a period of 14 years, 25 sites known to have been previously contaminated with *B. anthracis*, were sampled and spore counts determined at multiple times and sampling positions. A total of 136 (58.6%) soil samples yielded *B. anthracis*. Genotype data generated from 10 of these isolates, as expected, confirmed that they were closely related with minimal diversity (Appendix A, [13]).

Sampling position was a good predictor for spore count (p<0.001). Although no contaminated site had a slope with a gradient > 15°, plotting the estimated, modelled means of spore count at different sampling positions demonstrated that counts were greater at a short distance down a slope from the centre (Position 1) of the contaminated area (Figure 1B). While we did not find any records of significant flooding at any of the sites studied, the Kars region is, however, subject to repeated sessions of freezing and thawing due to the melting of winter snow [14], and as a consequence the water table level is likely to fluctuate.

Although there were notable variations in spore counts between sites (Figure 2), statistical modelling was unable to find any association between the degree of contamination at a site and its location at either Village or District level (p=0.995 and p=0.298, respectively).

While there were no reported cases of human infection associated with living in close proximity to spore contaminated sites [7], levels of >10^5^ spores per gram of soil have been estimated to be a lethal dose for cattle grazing in Etosha [15]. Whether this is the same for cattle in the Kars region of eastern Türkiye has yet to be determined.

In contrast to studies in other parts of the world, there was no evidence that spore counts were linked to the time interval between contamination and the time of sampling (Figure 3). The effect of time was considered while holding the sampling position constant throughout the statistical modelling and this further confirmed a lack of effect (p=0.335), suggesting that the spores retained their viability throughout the time period studied.

While a main time effect was not obvious, it is clear that there is considerable variation in spore count. Two sites (F in Village γ, Kars District; R in Village τ, Selim District) in particular, were sampled repeatedly over time across a range of positions, enabling assessment of the variability of viable spore detection over time at a specific position. Figure 4 is a graphical representation of spore counts found at different time intervals in soil samples from 16 of the 25 sites studied.

On 21 occasions, spores were not isolated at one time point but were subsequently detected at the same position at a later date; these samples are highlighted yellow in Appendix A and the changing spore counts over time for 16 of the sampling positions involved are shown in Appendix A. At seven of these positions, samples fluctuated from positive through negative and then to positive for *B. anthracis* spores once more, and for a further four positions, they fluctuated repeatedly between negative and positive at different time points (Appendix A).

## 4. Discussion

The study reported in this paper provides background information to a recently published clinical study of the anthrax toxin specific immune response of individuals living in *B. anthracis* spore contaminated areas of the Kars region of eastern Türkiye [7]. The clinical study identified individuals with antibodies to toxin components suggestive of subclinical exposure to the pathogen. As the nature of this exposure was unknown, we sought to characterise the background level of environmental contamination by monitoring spore levels at known contaminated sites.

Locations known to be contaminated with *B. anthracis* spores were repeatedly sampled for the presence of the pathogen in some cases over a 14-year time frame. While there were notable variations in spore levels recovered at different times and sites, statistical modelling did not identify any association between the degree of contamination and geographic location. This suggests that the sites studied did not differ significantly in factors that affect spore survival.

While there was no statistical association between spore numbers and specific locations in our study, the position at which samples were taken from a contaminated site was a good predictor for recovery of the pathogen. In a study of spore contaminated sites in Canada, only 28.4% of samples taken from locations where the soil had been saturated with body fluids escaping the carcass were positive for *B. anthracis* spores [8]. We also found that recovery of *B. anthracis* spores from a specific sampling point varied with time in that 21 previously negative sampling positions yielded positive results upon retesting at a later date. A similar switch from negative to positive was also seen in a study of spore contaminated sites in Etosha National Park in Namibia [9,10]. The reasons for these fluctuations in spore recovery are unclear, but highlight the importance of collecting multiple samples in different positions over extended periods of time from a contaminated site, so as not to miss the presence of the pathogen.

In contrast to studies in other parts of the world, there was no evidence of a reduction in spore viability over the time course of the study. Indeed, spore concentrations of >10^2^/g soil were found in 10 (66.7%) of the 15 samples taken 10 or more years following initial contamination. We also observed no difference between sites where animal carcasses had been butchered on open ground and where they had been buried, highlighting the importance of appropriate decontamination, as burial on its own does not eliminate spores from the environment.

These results differ from those reported from a series of studies in Etosha which observed that *B. anthracis* spore numbers in soil declined exponentially over time with numbers as high as 1.19 × 10^9^ spores/g of soil reducing to no detectable spores over a 10-year period [15,16,17]. However, in some natural conditions, spores are known to persist for decades, with spores of the Vollum strain of *B. anthracis* being detected more than 40 years after soils were experimentally inoculated on Gruinard Island off the coast of Scotland [18]. Similarly, re-emergence of anthrax in reindeer in the Yamal region of Siberia in 2016 more than 70 years after the last known case, together with sporadic cases originating from unknown environmental sources in Sweden, strongly suggests that persistence times can exceed several decades under certain conditions [19,20].

Whilst the factors affecting persistence of spore reservoirs in soil are not fully understood, it is possible that changes in local weather patterns due to global warming could contribute to an increase in spore numbers and a corresponding increase in outbreaks [21]. Melting of the permafrost combined with changes in rainfall patterns [22] may result in an increase in local flooding that brings trapped spores to the surface from previously undisturbed reservoirs, replenishing any spores that may have died [23].

The rate at which a spore pool decays is also likely to depend on local environmental conditions; for example, larger concentrations of spores are found in soils having slightly alkaline pH (>6.0), higher organic matter and higher calcium content [24]. Features of the exosporium have also been shown to affect the ability of *B. anthracis* spores to bind to different soil types [25]. Spores persist best in dry soils where microbial activity is minimal. In moist soils viability is usually in the range of 3 months to 3–4 years, but rarely longer [24].

While an association has been made between the level of environmental contamination and infectivity in grazing cattle in Etosha, with >10^5^ spores/gram of soil being identified as a lethal dose [15], the same cannot be said for humans. Cases of clinically confirmed anthrax infection arising from direct contact with contaminated soil are extremely rare [26]. A Daily Telegraph report [27] in 2023, of Russian troops contracting anthrax as a consequence of digging trenches in the Zaporizhzhia region of Ukraine, appears at odds with a review by Finke et al. [28] which concluded that “there is no scientific evidence proving for soil-borne anthrax in military animals and soldiers even in case of intensive exposure during heavy disturbance of soil structure in known endemic areas”. Interestingly, these authors go on to say that in highly endemic areas there may be a risk of infection due to contact with soil from recent burial sites of infected animals. Ukraine has 10,000 known anthrax burial sites and it is estimated that there are a further 6000 unregistered sites [29]. In our study, we determined the level of *B. anthracis* spores in soil collected from known contaminated locations cross the Kars region of Türkiye, some in close proximity to sites of human occupation. While spore numbers at some sites exceeded those predicted in experimental models to infect humans, the lack of recorded clinical cases of infection associated with environmental exposure suggests that infections, if they do occur, are extremely rare [30].

Animal studies have shown that grazing animals in Etosha develop antibody responses to the *B. anthracis* toxin, suggesting exposure to the pathogen at levels insufficient to cause an active infection [31,32]. A recent study of humans in the Kars region also detected toxin specific antibody responses in individuals with no history of clinical infection suggesting that they had also been exposed to low levels of the pathogen [7]. Whether this exposure was a consequence of contact with contaminated animals and their products or spore contaminated environments is unknown.

The ability to live and work in spore contaminated environments is of particular interest to those tasked with rendering an area safe for human occupation following the release of large numbers of *B. anthracis* spores [33]. The limited studies which are available, including our own, suggest that healthy humans can tolerate low level spore exposure without ill effects [3,4,5,6,7,34]. Further work is needed to identify what constitutes a “low level of exposure” given that a community is likely to contain individuals who vary in their susceptibility to the pathogen [35].

## 5. Study Limitations

This was an observational study, reporting common features of all samples received by the Veterinary Microbiology Department of Kafkas University in Kars during the period 2009–2023, that proved positive for viable *Bacillus anthracis* spores. Therefore, the results do not give an indication of the prevalence or distribution of anthrax-contaminated sites in the Kars region. Rather, this study describes common features of samples from known contaminated sites, emphasises the longevity of infectious anthrax spores in soil, and underlines the importance of thorough decontamination and appropriate disposal of infected animal carcasses and tissues.

## Figures and Tables

**Figure 1 microorganisms-12-01944-f001:**
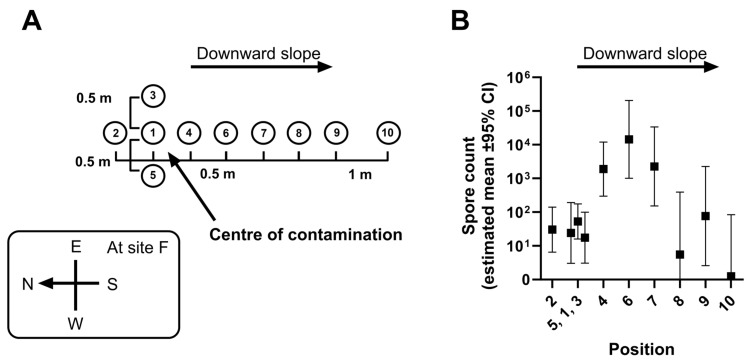
**Spore counts in samples from different positions at contaminated sites in the Kars region.** Panel (**A**): Standardised site sampling diagram: Position 1 was at the centre-point of the contaminated area. Panel (**B**): Estimated means (±95% CI) of spore counts at sampling positions 1–10 derived from a mixed model analysis of data from all the sites studied. The output of the mixed model analysis is shown in Appendix A and the estimated means (±95% CI) of spore counts are listed in Appendix A.

**Figure 2 microorganisms-12-01944-f002:**
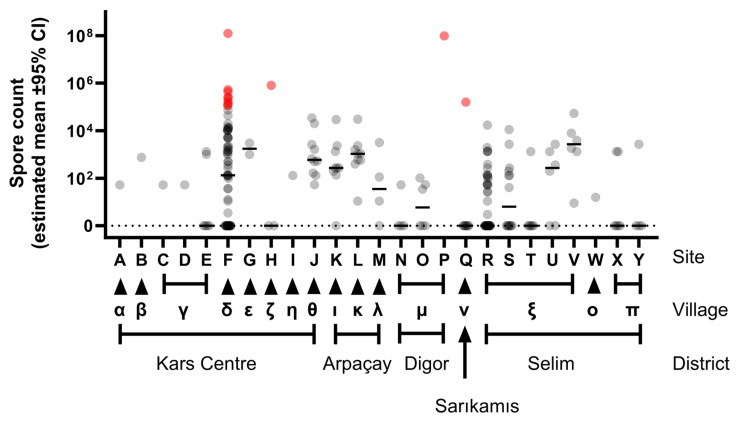
**Spore counts at multiple sites across the Kars region of Türkiye.** Counts are shown as a data point for each sample with the median line added. The points are partially transparent to enable data density to be visualised; thus darker shades of grey show spore count values where data points overlap. Samples were taken at one or more times and/or positions relative to the centre-point of each contaminated site (coded A to Y). The x-axis shows the closest Village (coded α to π) to the site and the administrative District within Kars province. Data points presented in red represent spore levels in excess of 10^5^ spores/gram of soil, which is a level estimated to be a lethal dose for grazing cattle [15].

**Figure 3 microorganisms-12-01944-f003:**
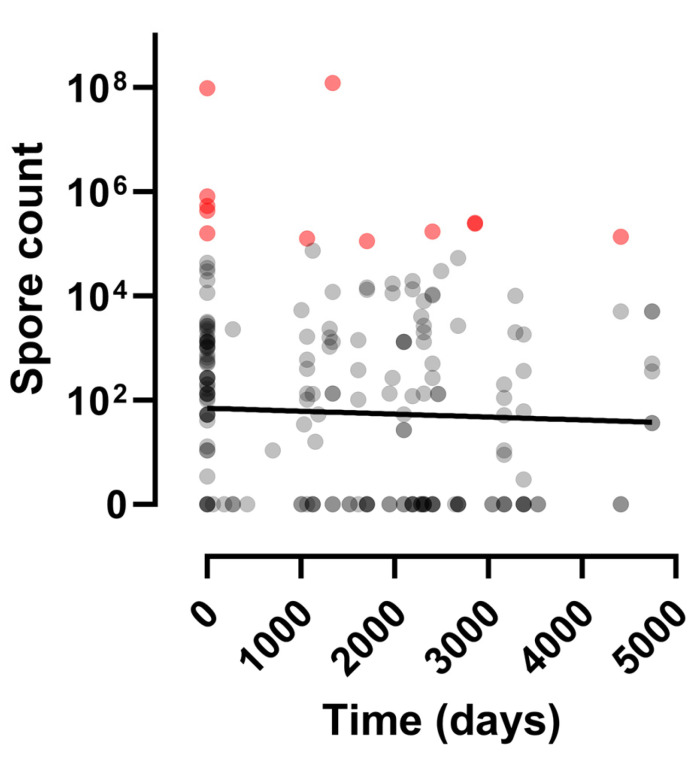
**A plot of *B. anthracis* spore yield relative to time since contamination at multiple sites across the Kars region.** This scatter plot ignores sampling position. A linear regression line has been added to illustrate the lack of time effect. Spore concentrations of >10^2^/g soil were found in 10 (66.7%) of the 15 samples taken 10–13 years following a contamination event. The points are partially transparent to enable data density to be visualised; thus darker shades of grey show spore count values where data points overlap. Data points presented in red represent spore levels in excess of 10^5^ spores/gram of soil, which is a level estimated to be a lethal dose for grazing cattle [15].

**Figure 4 microorganisms-12-01944-f004:**
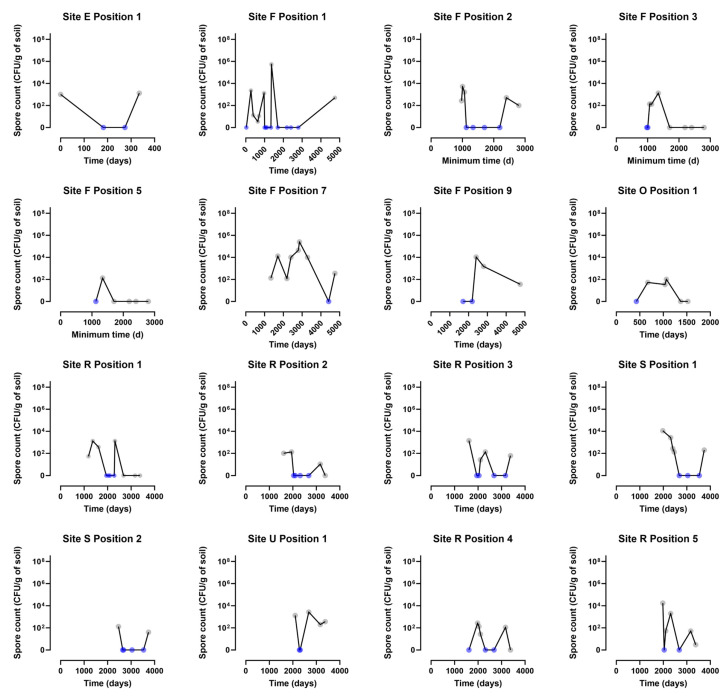
**Plots of *B. anthracis* spore counts relative to time since contamination, at 16 of the 25 sites studied.** The panels show the individual sites that were monitored across repeated time points. The line on these individual plots shows the geometric mean. Samples fluctuated between positive and negative.

## Data Availability

The original contributions presented in the study are included in the article/Appendix A, further inquiries can be directed to the corresponding author.

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
