# Peer review of "Soil Sample Analysis of Bacillus anthracis Contaminated Animal Burial Sites"

_microorganisms, 2024, doi:10.3390/microorganisms12101944_

Round 1
Reviewer 1 Report
Comments and Suggestions for Authors
In manuscript «Soil sample analysis of Bacillus anthracis contaminated animal burial sites in the Kars region of eastern Türkiye» considers the viability of anthrax spores in soil. The manuscript is in line with the theme of the journal. The article is a relevant study and the findings are interpreted correctly. The conclusions are consistent with the evidence and arguments presented. The manuscript complements existing knowledge, the research is clear and well-structured for the readership.1. The article presents relevant research, has novelty and originality in its content.
2. Finalize the Abstract: provide a clear purpose of the work and its uniqueness.
3. Format your article according to the journal's rules.
4. The conclusions are consistent with the evidence and arguments presented.
Author Response
Thank you for your kind general comments about our paper.
I have corrected the formatting irregularities that you identified and have attached a copy of the paper with all the changes made in response to the three Reviewer's comments.
With best wishes
Hugh Dyson

Reviewer 2 Report
Comments and Suggestions for Authors
Dear Authors,
The manuscript titled "Soil sample analysis of Bacillus anthracis contaminated animal burial sites in the Kars region of eastern Türkiye,' submitted by Sahin et al. (2024), on microorganisms aimed to evaluate 232 soil samples collected during 2009- 2023 from sites known to be contaminated with B. anthracis spores following burial or butchering of infected animal carcasses. This is an extremely important issue, as B. anthracis is a bacterium that causes serious problems and can be transmitted to humans primarily through close contact with the meat and products of infected animals. Upon reviewing the manuscript, I found that the text has no clarity and organizational issues, making it confusing. I recommend updating the cited literature. The methodology has serious flaws, preventing the experiment from being reproducible. Additionally, there is no map showing the study areas, and the standardization of sample collections is not clearly defined. The graphs, particularly figure 04, are of poor quality. Therefore, my recommendation is to reject the manuscript in its current form.
Comments on the Quality of English LanguageImprove it
Author Response
Many thanks for your comments.
We have made the modifications you suggested with regard to the clarity of the two figures with multiple panels and hope you will agree that this enhances the quality of the paper. We have changed the way way we described the soil sampling methodology we used, and believe that it should now be easily reproducible. We have also added three more recent references [21-23] to th Discussion as you suggested.
Given the fact that three of the authors are native English speakers, with long working experience of writing scientific English, we do not believe that submitting the manuscript to an external company for linguistic review is necessary.
With best wishes
Hugh Dyson
Reviewer 3 Report
Comments and Suggestions for Authors
Dear Authors,
The manuscript has a good significant of content and interest to read about the longevity of viable anthrax spores in soil in agricultural environments following decomposition of infected animal carcasses which need prolong bacteriological monitoring. Authors should make further studies by how create the appropriate way for decontamination these sites and eliminate these prolong living spores.
Averall, the manuscript has a good immpresion for being published

Author Response
Thank you for your kind comments about our paper.
We have amended the sentence in the Abstract that you pointed out was unclear and trust that you will find the changes satisfactory.
With best wishes
Hugh Dyson